# DAIR: Disentangled Attention Intrinsic Regularization for Safe and Efficient Bimanual Manipulation

## Abstract

We address the problem of safely solving complex bimanual robot manipulation tasks with sparse rewards. Such challenging tasks can be decomposed into sub-tasks that are accomplishable by different robots concurrently or sequentially for better efficiency. While previous reinforcement learning approaches primarily focus on modeling the compositionality of sub-tasks, two fundamental issues are largely ignored particularly when learning cooperative strategies for two robots: (i) *domination*, *i.e.*, one robot may try to solve a task by itself and leaves the other idle; (ii) *conflict*, *i.e.*, one robot can interrupt another's workspace when executing different sub-tasks simultaneously, which leads to unsafe collisions. To tackle these two issues, we propose a novel technique called *disentangled attention*, which provides an intrinsic regularization for two robots to focus on separate sub-tasks and objects. We evaluate our method on five bimanual manipulation tasks. Experimental results show that our proposed intrinsic regularization successfully avoids domination and reduces conflicts for the policies, which leads to significantly more efficient and safer cooperative strategies than all the baselines. Our project page with videos is at https://bimanual-attention.github.io/.

## 1 Introduction

Consider the bimanual robot manipulation tasks such as rearranging multiple objects to their target locations in Figure 1 (a). This complex and compositional task is very challenging as the agents will first need to reduce it to several sub-tasks (pushing or grasping each object), and then the two agents will need to figure out how to allocate each sub-task to each other (which object each robot should operate on) for better collaboration. Importantly, two robots should avoid collision in a narrow space for safety concerns. While training a single RL agent that can solve such compositional tasks has caught research attention recently (Chang et al., 2019; Peng et al., 2019; Devin et al., 2019; Jiang et al., 2019; Li et al., 2021; 2020), there are still two main challenges that are barely touched when it comes to tackle bimanual manipulation: (i) *domination*, *i.e.*, one robot may tend to solve all the sub-tasks while the other robot remains idle, which hurts the task solving efficiency; (ii) *conflict*, *i.e.*, two robots may try to solve the same sub-task simultaneously, which result in unsafe conflicts and interruptions on shared workspace.

One possible solution is to design a task-allocation reward function to encourage better coordination. However, it is particularly non-trivial and often sub-optimal to manually design such a reward function for complex problems that contain a large continuous sub-task space, such as the rearrangement task in Figure 1 (a). Moreover, even with the reward function described above in hand, it remains unclear how to reduce collisions, particularly for the tasks that require two robots to act simultaneously and safely. For example, in the task shown in Figure 1 (d), one robot needs to push the green door to make space for the other robot to move the blue box to the goal position. However, these two robots can easily interrupt and collide with each other when they perform these coordination actions.

We consider an alternative setting using sparse rewards without explicitly assigning sub-tasks to the robots. However, this leads to another challenge: How to encourage the robots to explore collaborative and safe behaviors with limited positive feedbacks? For bimanual manipulation, an intrinsic motivation is introduced by Chitnis et al. (2020b), leveraging the difference between the

Figure 1: Five bimanual manipulation tasks. (a) Rearrange the blocks to goal positions. (b) Stack the blocks into a tower. (c) Open the box and put the block inside. (d) Open the green door and push the block through the wall to the goal position. There are springs on both the box in (c) and the door in (d), which will close automatically without external force. Thus it requires one robot to hold the box cover and the door. (e) Lift up and rotate a bar with two arms to a target configuration, where the gripper is locked as closed, so it cannot be done by one arm. More details about our environments are in Appendix A.

actual effect of an action (taken by two robots) and the composition of individual predicted effect from each agent using a forward model. While this intrinsic reward encourages the two robots to collaborate for tasks that are hard to achieve by a single robot, it does not address the *domination* and *conflict* problems for efficient and safe manipulation.

In this paper, we present **DAIR**: **D**isentangled **A**ttention **I**ntrinsic **R**egularization which encourages the two robots to safely and efficiently collaborate on different sub-tasks during bimanual manipulation. Instead of designing a new intrinsic reward function, we introduce a simple regularization term for representation learning, which encourages the robots to *attend* to different interaction regions. Specifically, we adopt the attention mechanism (Vaswani et al., 2017) in both our policy and value networks, where we compute the dot-product between each robot representation and the object interaction region representations to obtain a probability distribution. Each robot has its own probability distribution to represent which interaction region it is focusing on. We define our intrinsic regularization as minimizing the dot product between the two probability distributions between two robots (*i.e.*, to be orthogonal) in each time step. By adding this loss function, different robots will be regularized to attend to different interaction points within their policy representation. This forces the policies to tackle sub-tasks over disjoint working space without interfering with each other. We remark that disentangled attention can be generalized to environments with multiple agents.

In our experiments, we focus on five diverse manipulation tasks in simulation environments with two Fetch robots as shown in Figure 1. These tasks not only require the robots to manipulate multiple objects (more than two, up to *eight*) with each object offering one interaction region, but also a single heavy object with multiple interaction regions (Figure 1 (e)). In our experiments, we show that our approach not only improves performance and sample efficiency in learning, but also helps avoiding the domination problem and largely reducing the conflicts for safe coordination between two robots. Moreover, the learned policies can also solve the task in fewer steps, which is the significance of bimanual cooperation compared to single-arm manipulation. We highlight our main contributions as:

- Observation for two important problems (*domination* and *conflict*) in training RL agents for safe bimanual manipulation, and a new robotics task set with one to eight objects.
- We propose DAIR, a novel and general intrinsic regularization. It not only improves the success rate in bimanual manipulation, solves the tasks more efficiently, but also reduces the conflicts between robots. This allows the robots to collaborate and coordinate more safely.

## 2 RELATED WORK

**Intrinsic motivation in reinforcement learning.** To train RL agents with sparse rewards, Schmidhuber (1991) first proposed to motivate the agent to reach state space giving a large model prediction error, which indicates the state is currently unexplored and unseen by the model. Such a mechanism is also called intrinsic motivation, which provides a reward for agents to explore what makes it curious (Oudeyer et al., 2007; Barto, 2013; Bellemare et al., 2016; Ostrovski et al., 2017; Huang et al., 2019). Recently, it has also been shown that the agents can explore with such an intrinsic motivation without extrinsic rewards (Pathak et al., 2017; Burda et al., 2018; 2019). Besides using prediction error, diverse skills can also be discovered by maximizing the mutual information between skills and states as the intrinsic motivation (Eysenbach et al., 2019; Sharma et al., 2020). While these approaches have achieved encouraging results in single agent cases, they are not directly applicable to environments with multiple agents. In our paper, we propose a novel intrinsic regularization for helping two robots work actively on different sub-tasks.

**Multi-agent collaboration.** Cooperative multi-agent reinforcement learning has exhibited progress over the recent years (Foerster et al., 2016; He et al., 2016; Peng et al., 2017; Lowe et al., 2017; Foerster et al., 2018; Sunehag et al., 2018; Rashid et al., 2018; Son et al., 2019; Wang et al., 2020a). For example, Lowe et al. (2017) proposed to extend the DDPG (Lillicrap et al., 2016) algorithm to the multi-agent setting with decentralized policies and centralized Q functions, which implicitly encourages the agents to cooperate. However, the problem of exploration still remains as a bottleneck, and in fact even more severe in multi-agent RL. Motivated by the previous success on a single agent, intrinsic motivation is also introduced to help multiple agents explore and collaborate (Foerster et al., 2016; Strouse et al., 2018; Hughes et al., 2018; Iqbal & Sha, 2019b; Jaques et al., 2019; Wang et al., 2020b). For example, Jaques et al. (2019) proposed to use social motivation to provide intrinsic rewards which model the influence of one agent on another agent's decision making. The work that is most related to ours is by Chitnis et al. (2020b) on the intrinsic motivation for synergistic behaviors, which encourages the robots to collaborate for a task that is hard to solve by a single robot. As this paper has not focused on the domination and conflict problems, our work on disentangled attention is a complementary technique to the previous work.

**Bimanual manipulation.** The field of bimanual manipulation has been long studied as a problem involving both hardware design and control (Raibert & Craig, 1981; Hsu, 1993; Xi et al., 1996; Smith et al., 2012). In recent years, researchers applied learning based approach to bimanual manipulation using imitation learning from demonstrations (Zollner et al.; Gribovskaya & Billard, 2008; Tung et al., 2020; Xie et al., 2020) and reinforcement learning (Kroemer et al., 2015; Amadio et al., 2019; Chitnis et al., 2020a;b; Ha et al., 2020). For example, Amadio et al. (2019) proposed to leverage probabilistic movement primitives from human demonstrations. Chitnis et al. (2020a) further introduced a high-level planning policy to combine a set of parameterized primitives to solve complex manipulation tasks. In contrast to these works, our approach does not assume access to pre-defined primitives. Both robots will learn how to perform each sub-task and how to collaborate without conflicts in an end-to-end manner, which makes the approach more general.

**Attention mechanism.** Our intrinsic motivation is built upon the attention mechanism which has been widely applied in natural language processing (Vaswani et al., 2017) and computer vision (Wang et al., 2018; Dosovitskiy et al., 2021). Recently, the attention mechanism is also utilized in multi-agent RL to model the communication and collaboration between agents (Zambaldi et al., 2018; Jiang & Lu, 2018; Malysheva et al., 2018; Iqbal & Sha, 2019a; Long et al., 2020). For example, Long et al. (2020) proposed to utilize attention to flexibly increase the number of agents and perform curriculum learning for large-scale multi-agent interactions. Li et al. (2020) adopt the attention mechanism to generalize multi-object stacking with a single arm. In our paper, instead of simply using attention for interaction among hand and a variable number of objects, we propose DAIR to encourage the agents to attend on different sub-tasks for better collaboration.

## 3 PRELIMINARIES

We consider a multi-agent Markov decision process (MDP) Littman (1994) with $N$ agents, which can be represented by $(S, A, P, R, H, \gamma)$. The state $s \in S$ and the action $a_i \in A$ for agent $i$ are continuous. $P(s^{t+1}|s^t, a_1^t, ..., a_N^t)$ represents the stochastic transition dynamics. $R_i(s^t, a_i^t)$ represents the reward function for agent $i$. $H$ is the horizon and $\gamma$ is the discount factor. The policy $\pi_{\theta_i}(a^t|s^t)$ for agent $i$ is parameterized by $\theta_i$. The goal is to learn multi-agent policies maximizing the return. In this paper, we tackle a two-agent collaboration problem ($N = 2$), but our method can generalize to more agents.

### 3.1 REINFORCEMENT LEARNING WITH SOFT ACTOR-CRITIC

We adopt the Soft Actor-Critic (SAC) Haarnoja et al. (2018) for reinforcement learning (RL) training in this paper. It is an off-policy RL method using the actor-critic framework. The soft Q-function for agent $i$ is $Q_{\theta_i}(s^t, a_i^t)$ parameterized by $\theta_i$. For agent $i$, there are three types of parameters to learn in SAC: (i) the policy parameters $\phi_i$; (ii) a temperature $\tau_i$; (iii) the soft Q-function parameters $\theta_i$. We can represent the policy optimization objective for agent $i$ as,

$$J_\pi(\phi_i) = \mathbb{E}_{s^t \sim \mathcal{D}}\left[\mathbb{E}_{a_i^t \sim \pi_{\phi_i}}[\tau_i \log \pi_{\phi_i}(a_i^t|s^t) - Q_{\theta_i}(s^t, a_i^t)]\right], \tag{1}$$

where $\tau_i$ is a learnable temperature coefficient for agent $i$, and $D$ is the replay buffer. It can be learned to maintain the entropy level of the policy:

$$J(\tau_i) = \mathbb{E}_{a_i^t \sim \pi_{\phi_i}}\left[-\tau_i \log \pi_{\phi_i}(a_i^t|s^t) - \tau_i \mathcal{H}\right], \tag{2}$$

where $\bar{\mathcal{H}}$ is a desired minimum expected entropy. The soft Q-function parameters $\theta_i$ for agent $i$ can be trained by minimizing the soft Bellman residual as,

$$J_Q(\theta_i) = \mathbb{E}_{(s^t, a_i^t) \sim \mathcal{D}} \left[ \frac{1}{2} (Q_{\theta_i}(s^t, a_i^t) - \hat{Q}(s^t, a_i^t))^2 \right], \qquad (3)$$

$$\hat{Q}(s^t, a_i^t) = R_i(s^t, a_i^t) + \gamma \mathbb{E} \left[ \max_{a_i^{t+1} \sim \pi_{\phi_i}} Q_{\theta_i}(s^{t+1}, a_i^{t+1}) \right]. \qquad (4)$$

Since we focus on collaborative robotics manipulation tasks, the reward is always shared and synchronized among the agents. That is, if one agent is able to finish a goal and obtain a reward, the other agents will receive the same reward.

## 3.2 CRITERIA FOR MEASURING EFFICIENCY AND SAFETY

Our core contribution is to ensure the manipulating efficiency and safety by reducing the problems of domination and conflict in the process of manipulation, which also improves the speed of finishing the task. We define three criteria which are all lower the better, and evaluate our approach using them in our experiments: (i) **Domination Rate:** We count how many steps an arm is interacting with an object in one episode as the manipulating steps. We compute the ratio of an agent's manipulating steps over the two agents' total manipulating steps. We select the maximum ratio as the Domination Rate. Ideally, we hope the Domination Rate to be close to $50\%$ which indicates both robots are actively interacting with the objects. (ii) **Conflict Rate:** It counts the percentage of the "conflict step" over all steps. We consider it a conflict step when the distance between two grippers is smaller than a small threshold. This means two robots are interrupting each other's action. (iii) **Finish Steps:** How many steps do two agents take to finish the task successfully. The maximum episode length in both environments is 100. Note that it's reasonable only when we consider all the metrics together, since robots can use trivial solution to reduce one of them, for example, only complete the task with one robot (extreme domination behavior) to avoid conflicts.

## 4 METHOD

Our goal is to design a model and introduce a novel intrinsic regularization to better train the policies for bimanual manipulation tasks. We hope the agents can learn to allocate the workload efficiently and safely. In this section, we will first introduce our base network architecture with the attention mechanism motivated by (Vaswani et al., 2017). Based on this architecture, we will then introduce DAIR and how to perform reinforcement learning with it.

### 4.1 NETWORK ARCHITECTURE

Policy and Q-function networks share the same architecture as follow. For simplicity, we omit superscript time $t$ when there is no ambiguity. We can then represent the state as $s = [s_1, \ldots, s_N, s_{N+1}, \ldots, s_{N+M}]$ where the first $N$ entities represent the state of the robot arms, the next $M$ entities represent the states of the interaction regions. Here we define an *interaction region* as a space on object that robot can manipulate on (grasp or push, etc) to handle the task. In Figure 1, for the first four tasks with smaller objects such as cubes, door, and cover, we define one interaction region located around each object. For the last task of adjusting a large heavy bar, we define it with two interaction regions with one region corresponding to one face. Note that the interaction regions can be specified according to the tasks flexibly, and our method can be generalized to any numbers of interaction regions.

For agent $i$, we have a set of *state encoder* functions $\{f_{i,1}(\cdot), \ldots, f_{i,N}(\cdot), f_{i,N+1}(\cdot), \ldots, f_{i,N+M}(\cdot)\}$ corresponding to the input states, as shown in Figure 2. Each state encoder function $f_{i,j}(\cdot)$ takes the state $s_j$ as the input and outputs a representation (512-D) for the state in our policy network. We use a 2-layer multilayer perceptron (MLP) to model $f_{i,j}(\cdot)$. While there are $N + M$ state encoder functions, there are only three sets of parameters (visualized by three different colors in Figure 2): (i) the parameters of the state encoder for agent $i$ itself $f_{i,i}(\cdot)$; (ii) the parameters of the other agents $f_{i,j}(\cdot), (1 \leq j \leq N, i \neq j)$ (shared); (iii) the parameters of all the interaction regions $f_{i,j}(\cdot), (N + 1 \leq j \leq N + M)$ (shared). In this way, our model can be extended to environments with different number of interaction regions and agents. We represent the policy for agent $i$ as,

$$\pi_{\phi_i}(a_i|s) = h_i(f_{i,i}(s_i) + \text{LayerNorm}(g_i(v_i))), \qquad (5)$$

where $g_i(\cdot)$ is one fully connected layer to further process the *attention embedding* $v_i$, which encodes the relationship between agent $i$ and all the state entities (including all agents and interaction regions). Adding attention embedding to $f_{i,i}(s_i)$ with a LayerNorm operator serves as a residual module to retain agent $i$'s own state information. The combined features are fed to a 2-layer MLP $h_i(\cdot)$. The output of $h_i(\cdot)$ is the action distribution. Note that the parameters of $h_i(\cdot), g_i(\cdot)$ are not shared across the agents.

Motivated by Vaswani et al. (2017), we further define the attention embedding $v_i$ for agent $i$ as,

$$v_i = \sum_{j=1}^{N+M} \alpha_{i,j} f_{i,j}(s_j), \quad \alpha_{i,j} = \frac{\exp(\beta_{i,j})}{\sum \exp(\beta_{i,j})}, \quad \beta_{i,j} = \frac{f_{i,i}^T(s_i) W_q^T W_k f_{i,j}(s_j)}{\sqrt{d_q}}, \quad (6)$$

where $W_q$ represents one fully connected layer to encode the *query* representation $f_{i,i}(s_i)$ and $W_q$ represents another fully connected layer to encode the *key* representation $f_{i,j}(s_j)$. $d_q$ is the dimension of the query representation. $\beta_{i,j}$ represents the correlation between agent $i$ and all the other entities.

It is then normalized by a softmax function to $\alpha_{i,j}$ as the probability value, which indicates where agent $i$ is "attending" or focusing on in the current time step and $\alpha_i \in \mathbb{R}^{N+M}$. $v_i$ is computed via a weighted sum over all the state encoder representations. For Q-function, there are two modifications: (i) The state encoder $f_{i,i}^Q(s_i, a_i)$ for the agent $i$ also takes in the action as inputs; (ii) The final layer of the Q-function network outputs a scalar value instead of an action distribution as $Q_{\theta_i}(s^t, a_i^t)$, which is used in Eq. 1 and Eq 3.

## 4.2 Disentangled Attention as Intrinsic Regularization

We propose *disentangled attention as intrinsic regularization (DAIR)* to improve the state encoder representations for solving the problem of *domination* of a single agent and the *conflict* between the agents. Each manipulation task can be decomposed to multiple sub-tasks, each with a different interaction regions. Our key insight is to encourage different robots to attend or focus on different interaction regions, and consequently to work on different sub-tasks.

Specifically, we look into the softmax probability $\alpha_{i,j} \in [0,1]$ from the attention mechanism in Eq. 6. This variable represents how much attention agent $i$ is putting on interaction region/agent $j$. To encourage the agents to focus on different entities, we propose the following loss function for agent $i$ as,

$$L_{\text{attn}}(\phi_i) = \sum_{j=1, j \neq i}^{N} < \alpha_i, \alpha_j >^2, \quad (7)$$

where $< \cdot, \cdot >$ denotes dot product of two vectors. This loss forces the dot product between two attention probability vector to be small, which encourages different agents to

Figure 2: Our model framework. We use attention mechanism to combine all embedded representations from agents and interaction regions. The output of attention module, together with another embedded vector from $s_i$ are summed together with $\oplus$. The combined feature is fed into a 2-layer MLP $h_i$ to output $a_i$. The intrinsic loss is computed from the attention probability $\alpha_i$ and encourages the agents to attend to different sub-tasks.

attend on different entities (including agents and interaction regions). We call this particular attention maps regulated by the orthogonal constraint as *disentangled attention*.

Recall that $\alpha_i$ is predicted via the state encoder functions, parameterized by a part of $\phi_i$. Instead of proposing a new reward function, our disentangled attention regularization is directly applied on learning the state encoder representation itself. The training objective for the policy network and

Q-function can be represented as,

$$\min_{\phi_i} J_\pi(\phi_i) + \lambda L_{\text{attn}}(\phi_i), \ \min_{\theta_i} J_Q(\theta_i) + \lambda L_{\text{attn}}(\theta_i). \tag{8}$$

where $\lambda = 0.05$ is a constant to balance the reinforcement learning objective and our regularization.

**Implementation Details.** The robot state $s_i(1 \leq i \leq N)$ contains the joint positions and velocities and the end-effector positions. Thus each robot can reason the other robot's joint state and avoid conflicts. Each interaction region $s_i(N + 1 \leq i \leq N + M)$ contains the target interacting position, velocity, pose and its goal position, which are all in $(x, y, z)$-coordinates. The action representation contains the positional control and the gripper motion information.

## 5 EXPERIMENTS

**Environment and setting.** We perform our experiments on complex bimanual manipulation tasks ($N = 2$) in the MuJoCo simulator (Todorov et al., 2012). By further leveraging curriculum learning, we successfully complete the scenarios with up to eight objects with sparse rewards. We evaluate the sample efficiency of training, conflict rate, domination rate, and completion steps across approaches to demonstrate that DAIR can (i) help discover efficient collaboration strategies; (ii) improve efficiency and safety by avoiding domination and conflict; (iii) bring adaptation capability with learned task decomposition knowledge; (iv) retain learning capability of synergistic skills.

**Baselines.** We compare our approach with three baselines: (i) The same architecture as our model with the attention mechanism, but without the intrinsic regularization (**Attention**); (ii) SAC with Multi-layer Perceptron (**MLP**) neural network; (iii) Multi-Agent Deep Deterministic Policy Gradient (Lowe et al., 2017) with MLP (**MADDPG + MLP**). We also tried replacing DDPG with SAC in MADDPG but observed minor differences. Thus we only report results with **MADDPG + MLP** for simplicity.

**Training details.** All networks and learnable parameters are trained with Adam optimizer (Kingma & Ba, 2015) with learning rate $0.0001, \beta_1 = 0.9, \beta_2 = 0.999$. We set the discount factor as $\gamma = 0.98$, buffer size as 1M, and batch size as 512 for all tasks. We follow the replay $k$ setting in HER (Andrychowicz et al., 2017), and set $k = 4$ with the future-replace strategy. We update the network parameters after every two episodes. The episode length equals 50 times the object number for each environment. We train all the methods with 3 seeds and report both the mean and standard derivation for the success rate. More details are in Appendix B.

### 5.1 SUB-TASK ALLOCATION FOR COLLABORATION

We first perform our experiments on two tasks that requires alternately operating different parts in a certain order to show DAIR balancedly and safely allocates the sub-tasks for solving one task. The first task is *Open Box and Place* (Figure 1 (c)): The robots need to put the blue block object inside the box with a sliding cover, which requires one robot arm to open the sliding cover for the other robot arm to put the object inside. The second task is *Push with Door* (Figure 1 (d)): The robots need to push the blue object to the goal on the other side of a sliding green door that requires one robot arm to open it and clear the way for the pushing arm (with the grasping function disabled). In both cases, we also apply a force on the sliding cover/door for it to bounce back to its original position if there is no outside forces. The following results show that DAIR not only helps achieve better sample efficiency and performance, but more importantly, addresses the problems of *domination* and *conflict*, thus further reduces the steps to finish the task at the same time.

**Reward setting.** We consider two different reward settings: (i) a *sparse* reward setting where the agents only obtain a reward 1.0 when the block is on the target position; (ii) a *informative* reward setting which gives a reward 1.0 when the box/door is open and another reward 1.0 when the block reaches to the goal. If the block reaches its goal in a trial, we count it as a successful trial.

**Comparison on success rate.** We plot the success rate of all the methods over the environment steps in Figure 3. We can observe that DAIR achieves better sample efficiency and better success rate than the baselines in most cases. We also observe that in both environments, DAIR and Attention achieve better success rate in the sparse reward setting than using informative reward. The reason is that sparse reward offers more flexibility for the agents to collaborate under the guidance with intrinsic disentangled attention, while the explicit informative reward can lead to local minimum more easily.

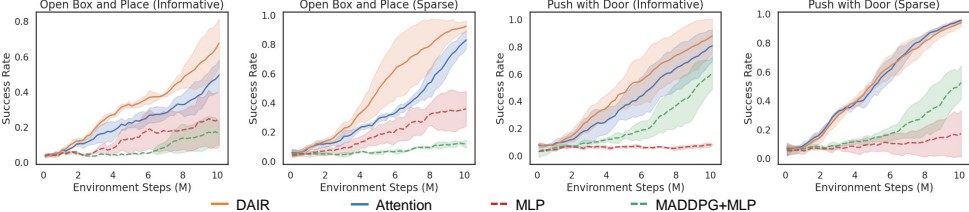

Figure 3: Performances of different methods on two bimanual manipulation tasks, *Open Box and Place* (2 on the left) and *Push with Door* (2 on the right). We consider two reward settings for each task, (i) a *sparse* reward (right in each group), where agents only receive a success reward when all the goals are reached; (ii) an *informative* reward (left in each group), where agent will additionally receive a reward for reaching each individual goal in addition to the final success reward.

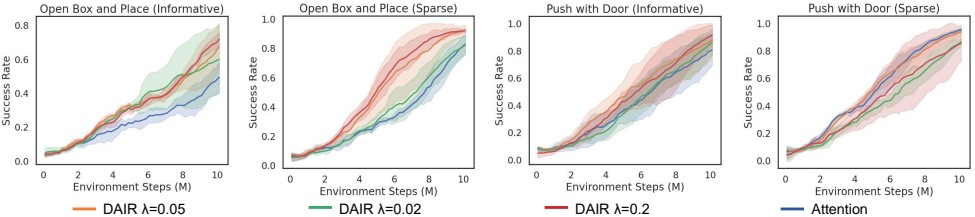

Figure 4: Ablation studies on the value of $\lambda$. Our method is generally robust to the choice of $\lambda$, when even is large (e.g., $\lambda$=0.2). In our practice, we choose $\lambda$=0.05 for all the experiments.

**Ablation on $\lambda$.** We set the hyperparameter $\lambda = 0.05$ (defined in Equation 8 for balancing the regularization) in all our experiments. To study the stability of DAIR, we perform ablation on different values of $\lambda$ in Figure 4. We observe that our method is robust to the change of $\lambda$ from $0.02$ to $0.2$.

**Comparison on the three criteria.** Table 1 shows the comparison on the three criteria defined in Section 3.2. We observe significant improvements over all the settings using intrinsic regularization, which proves that using disentangled attention can lead to better collaboration. For example, in the task of Open Box and Place with informative reward, our approach achieves almost half less conflict rate, $24\%$ less domination rate, and 12 fewer steps comparing to the Attention baseline without the intrinsic regularization. For Push with Door using sparse reward, we reduce more than half the conflict rate.

We perform ablation by introducing an extra collision penalty during training: Two robots will receive $-1.0$ reward if their grippers collide to each other. Note that such a reward is *not realistic* in practice since we do not hope the robots to collide to get the reward. We show the *Conflict rate* for both Attention baseline and our approach training with this collision penalty in Table 2. DAIR shows consistent improvements and remains to be an effective way to reduce conflicts even with the collision penalty.

**Visualization on attention probability $\alpha$.** We visualize the two tasks in Figure 5. In each task, we visualize the attention $\alpha_1$ and $\alpha_2$ for each robot in two rows. Each attention vector $\alpha_i$ contains four items that correspond to left arm (1st column), right arm (2nd column), and the two task-specific interaction regions (last 2 columns). Figure 5 (a) shows the Push with Door task: The left arm is interacting with the object block, so it has a high value in the corresponding probability $\alpha_{1,3}$ (1st row and 3rd

Table 1: Conflict rate (%), domination rate (%) and average finishing steps of our method and the baseline with pure attention on different tasks. Lower value is better. Box: Open Box and Place. Door: Push with Door.

| Domination Rate | Attention | DAIR |
|---|---|---|
| Box (Informative) | $77.2_{\pm 2.9}$ | $\mathbf{53.4_{\pm 0.5}}$ |
| Box (Sparse) | $74.5_{\pm 2.8}$ | $\mathbf{62.6_{\pm 6.4}}$ |
| Door (Informative) | $83.7_{\pm 4.8}$ | $\mathbf{76.5_{\pm 5.5}}$ |
| Door (Sparse) | $68.8_{\pm 6.7}$ | $\mathbf{66.9_{\pm 7.0}}$ |

| Conflict Rate | Attention | DAIR |
|---|---|---|
| Box (Informative) | $7.4_{\pm 0.8}$ | $\mathbf{4.0_{\pm 2.1}}$ |
| Box (Sparse) | $6.7_{\pm 5.0}$ | $\mathbf{3.6_{\pm 2.3}}$ |
| Door (Informative) | $35.3_{\pm 19.0}$ | $\mathbf{23.3_{\pm 16.6}}$ |
| Door (Sparse) | $44.1_{\pm 15.1}$ | $\mathbf{18.7_{\pm 11.7}}$ |

| Finish Steps | Attention | DAIR |
|---|---|---|
| Box (Informative) | $33.6_{\pm 5.5}$ | $\mathbf{21.3_{\pm 3.2}}$ |
| Box (Sparse) | $\mathbf{39.2_{\pm 9.8}}$ | $40.0_{\pm 11.4}$ |
| Door (Informative) | $23.0_{\pm 4.4}$ | $\mathbf{22.8_{\pm 5.7}}$ |
| Door (Sparse) | $30.3_{\pm 8.0}$ | $\mathbf{23.3_{\pm 6.6}}$ |

Table 2: Conflict rate (%) of our method and the attention baseline on different tasks with *Collision Penalty*. Lower value is better. Box: Open Box and Place. Door: Push with Door.

| Conflict Rate | Attention | DAIR |
|---|---|---|
| Box (Informative) | $10.4_{\pm 5.6}$ | $\mathbf{3.9_{\pm 1.8}}$ |
| Box (Sparse) | $3.5_{\pm 2.1}$ | $\mathbf{2.3_{\pm 0.9}}$ |
| Door (Informative) | $5.3_{\pm 1.19}$ | $\mathbf{3.4_{\pm 1.9}}$ |
| Door (Sparse) | $12.2_{\pm 3.0}$ | $\mathbf{4.7_{\pm 1.1}}$ |

Table 3: Success rate (%) on *Stack Tower* of different methods for each curriculum learning and adaptation stage. $a \rightarrow b$ means adapting the policy trained on $a$ objects to $b$ objects. *2 towers* means the agents need to stack two separate towers.

| #object | 1 | 2 | 3 | 2→3 | 3→4 | 2→4 (2 towers) |
|---|---|---|---|---|---|---|
| DAIR | **100**±**0.0** | **98.9**±**0.8** | **68.3**±**8.5** | **53.3**±**12.5** | **23.3**±**4.7** | **17.5**±**4.3** |
| Attention | 98.7±0.9 | 96.3±0.5 | 42.0±8.3 | 41.3±9.8 | 3.3±4.7 | 0.0±0.0 |

Table 4: Success rate (%) on *Rearrange* of different methods for each curriculum learning and adaptation stage. $a \rightarrow b$ means adapting the policy trained on $a$ objects to $b$ objects.

| #object | 1 | 2 | 3 | 2→3 | 3→4 | 2→4 | 3→8 |
|---|---|---|---|---|---|---|---|
| DAIR | **96.7**±**3.4** | **98.9**±**0.8** | **89.0**±**1.4** | **74.3**±**5.8** | **64.3**±**4.2** | **53.0**±**9.4** | **33.3**±**12.5** |
| Attention | 91.0±6.2 | 90.7±0.5 | 66.7±3.3 | 46.5±3.5 | 3.3±4.7 | 3.3±4.7 | 0.0±0.0 |

column); the right arm is interacting with the door, it also has a high value in the corresponding probability $\alpha_{2,4}$ (2nd row and 4th column). Similarly in Figure 5 (b) for the Open Box and Place task, a high probability with $\alpha_{i,j}$ indicates the $i$th arm is interacting with object $j$. The two interaction regions here are the block object (3rd column) and the box cover (4th column).

## 5.2 GENERALIZING TO MORE OBJECTS WITH CURRICULUM LEARNING

We increase the number of objects and train with curriculum to show that the regularized representation gains better learning capacity when task gradually becomes harder. We test on *Stack Tower* (Figure 1 (b)), where the robots need to stack objects as a tower with indicated goal positions; and *Rearrangement* (Figure 1 (a)), where the robots need to rearrange the objects to their own goal locations on the table. When manipulating one object in these environments, it is easy for the arm to perturb other objects without intention. We train RL agents for both tasks in the *informative reward* setting: the agents will receive a reward 1.0 when each object reaches its goal. We leverage curriculum learning to start training the agents to manipulate one object and then gradually increase the objects to three to the end.

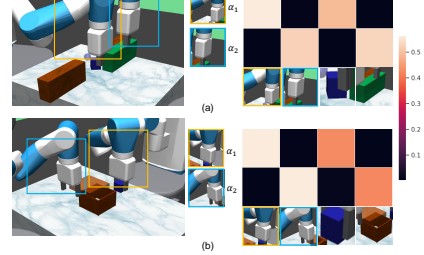

Figure 5: Visualization of attention $\alpha_i$. Each row corresponds to one robot arm attending to four items. (a) *Push with Door*: one robot holds the door while the other pushes the block; (b) *Open Box and Place*: one robot opens the box while the other picks the block.

We evaluate our approach on two aspects: (i) How does the approach perform in each curriculum stage; (ii) How does the approach generalize to object numbers that exceed its training number (up to **eight** objects in the Rearrangement task). DAIR achieves better results in both aspects, especially in generalization to multiple objects.

**Results on each curriculum stage.** We first compare DAIR to Attention on 3-block Rearrangement and Stack Tower tasks. Note that both **MLP** and **MADDPG+MLP** baselines cannot handle a flexible number of objects due to fixed dimensions of inputs. Thus it is not applicable in these two tasks with curriculum learning, and directly training both of them with 3 objects leads to zero success rate. This also suggests that DAIR and Attention have the flexibility to handle a variant number of input objects. DAIR gains significant improvements over Attention in all different training stages. Our gain over Attention even becomes larger as the number of objects increases.

**Results on generalization.** We conduct generalization experiments on both tasks where we test the policies trained with $i$ objects on the same environment with $i + k$ objects. We show the results of generalization success rate in Table 3 and 4 with the columns labeled by $i \rightarrow i+k$. For Stack Tower, DAIR trained with 2-block stacking generalizes to stacking 2 towers each with 2 blocks (last column in Table 3), while Attention completely fails. For Rearrangement, DAIR can rearrange 8 objects even we only train it to rearrange 3 objects (last column in Table 4), while Attention fails to generalize to even 4 objects. We conjecture that the reason for improvement is similar to conclusions in continual learning area (Delange et al., 2021): regularizing parameters with simple $L2$ loss avoids overfit to specific stage of task and gains better generalization ability in continually shifted tasks.

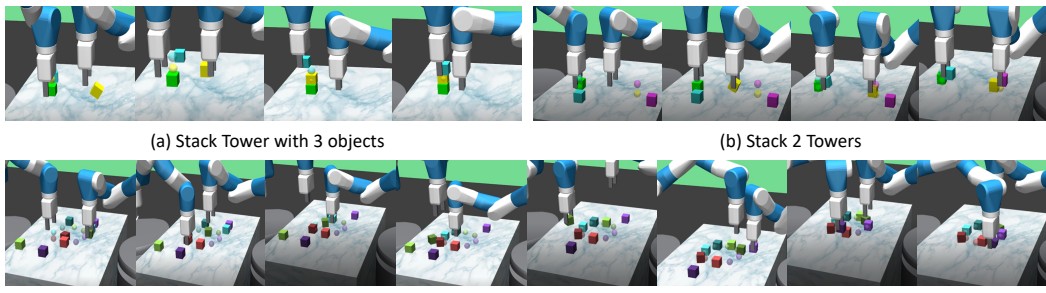

(a) Stack Tower with 3 objects             (b) Stack 2 Towers

(c) Rearrangement with 8 objects

Figure 6: Visualization of bimanual manipulation. For each object, we represents its goal as a transparent dot in the same color. (a) Both arms are picking up objects and alternatively stacking them into a tower; (b) To stack two tower, each arm is working on one tower that is close to it; (c) We show the two arms can collaborate without conflict to pick up the 8 objects to their target locations.

**Visualization on stacking and rearrangement.** We visualize the three demonstrations for our approach in Figure 6: (a) stacking 3 blocks; (b) stacking 2 towers each with 2 blocks using the policy trained with 2 block-stacking; (c) rearranging 8 blocks to their target positions using the rearrangement policy trained with 3 blocks. For stacking tasks, both robots are able to pick up different objects without interrupting the other robot and the stacked tower. For the rearrangement task, the policy is transferred to rearrange 8 objects, far beyond the training object number 3. The two robots are still able to collaborate without conflicts to solve the tasks. Please refer to our project page for more policy visualization in videos.

## 5.3 SYNERGISTIC BEHAVIORS DISCOVERY

Besides manipulation tasks with multiple objects, DAIR retains the ability to help two robots collaboratively manipulate one object with multiple interaction regions. To analyze the ability on such synergistic skills learning, we conduct experiment on the *Adjust Bar* task, as visualized in Figure 1 (e). This task requires the two robots to lift and rotate a heavy bar to target height and orientation. We lock the gripper to the closed state and set the bar's mass and size large so the two robots have to collaborate to finish the task. The interaction region state inputs are represented by the 3D position of two sides of the bar. The goal is defined by the target positions of the two sides of the bar. The reward is sparse that only an accurate adjustment gives 1.0 to two robots. DAIR maintains advantage on performance over the Attention baseline as shown in Table 5 and Figure 7. Two robots can successfully clamp up the bar with the opposite forces with very consistent movements. Such results suggest that DAIR is not harmful for learning synergistic skills when it encourages the robots to look at different interaction regions. DAIR also serves as a mechanism to avoid overfitting to domination and help robots to finish the task efficiently with smaller finish steps.

Table 5: Adjust Bar task. DAIR improves baseline over domination rate and finish steps. Conflict rate is not computed here since two end effectors will be close when the task is solved.

|  | Attention | DAIR |
|---|---|---|
| Domination Rate | $56.1_{\pm 5.3}$ | $\mathbf{52.6_{\pm 0.8}}$ |
| Finish Steps | $23.5_{\pm 6.0}$ | $\mathbf{18.2_{\pm 8.6}}$ |

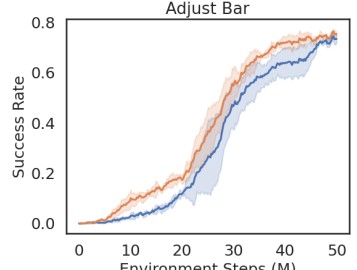

Figure 7: Training Curves of success rate. DAIR still retains learning efficiency though augmented with disentangled attention.

## 6 CONCLUSION

While previous works consider how to learn collaborative skills like synergistic behavior, we notice two main limitations in complex bimanual manipulation tasks: *domination* and *conflict*, corresponding to the efficiency and safety of control. We propose a simple and effective DAIR to solve these problems. We validate our approach on challenging bimanual manipulation tasks with multiple objects (up to 8 objects) or one objects with multiple interaction regions. We demonstrate that DAIR not only reduces the domination and conflict problems but also improves the generalization ability of the policies to manipulate much more objects than in the training environments. We hope our work contributes as a step towards safe robotics.

## 7 REPRODUCIBILITY STATEMENT

To ensure the reproducibility of our work, we provide the following illustrations in our paper and appendix:

- **Environment**: We provide the detailed description of the environment in Appendix A.
- **Evaluation Criteria**: We provide the detailed evaluation criteria for domination, conflict and finsh steps in Section 3.2.
- **Implementation Details**: We provide all implementation details and related hyperparameters in the end of Section 4.2, the beginning of 5 and Appendix B.

We are committed to releasing the code for our approach, the baselines, and the simulation environment. We believe the open source of our code, the task sets, and evaluation code will be an important contribution to the robotics community. We have released our videos in project page: https://bimanual-attention.github.io/, and we will release the code and environment on the same website upon publication.

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

# A  TASK DESCRIPTIONS AND DETAILS

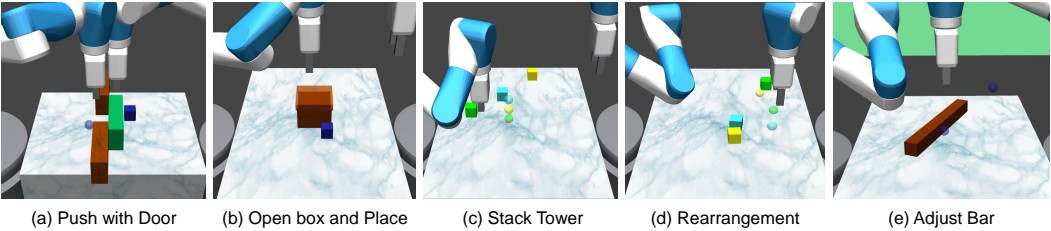

|   |   |   |   |   |
|---|---|---|---|---|
| (a) Push with Door | (b) Open box and Place | (c) Stack Tower | (d) Rearrangement | (e) Adjust Bar |

Figure 8: The environments used in our experiments

## A.1  ENVIRONMENT DESCRIPTIONS

**Push with Door, Figure 8(a).**   The two robots are placed on both sides of a 100cm × 70cm table, opposite each other (all robot manipulation environments are same for this setting). The goal is to push a block through a sliding door and make it reach the target position on the other side of the door. We put a spring on the sliding door, such that it will close automatically in the absence of external force. The initial positions of the projections of the two grippers onto the table plane are sampled in a 40cm × 40cm square on the table (all robot manipulation environments are same for this setting). The initial position of block and the goal position are sampled from a circle with radius 20cm around the table center. We fix the initial height of the two grippers (all robot manipulation environments are same for this setting), and set the initial position of door at the center of the table.

**Open box and Place, Figure 8(b).**   The task is to pick up the block on the table and place it into the box in table center. We put a spring on the sliding lid of the box, so it will close automatically in the absence of external force. The initial positions of the block and goal are sampled from a circle with radius 20cm at the center of the table, outside the box.

**Stack Tower, Figure 8(c).**   The task is to stack several blocks into a tower. All blocks are randomly sampled from a circle with radius 20cm around the center of the table. We perform curriculum learning in this environment, with one more block sampled in each stage. In the first stage in the curriculum, we have one block, we sample the corresponding goal with the height randomly from 0cm to 30cm. In each following stage, we sample one more object block and goal. After the first stage, the goals will form into a tower.

**Rearrangement, Figure 8(d).**   The task is to push multiple blocks to their corresponding target positions on the table. We perform curriculum learning in this environment, with one more block sampled in each stage. All blocks and goals are randomly sampled from a circle with radius 20cm around the table center. We train our method up to 3 blocks (3 curriculum stages) and generalize the approach to up to 8 blocks.

**Adjust Bar, Figure 8(e).**   The task is to adjust a heavy bar to the state that the two sides of it match with the two blue goal positions. That requires the height and the orientation are matched. We randomly sample the two goal positions but make sure the distance between them equals to the length of bar. We fix the gripper to force two robots synergistically use the force in opposite direction to pinch up the bar.

## A.2  OBSERVATION SPACE

The observation vector consists of object states, robot states, and the goals for the objects. Specifically, the object states consist of the position and velocity of all the objects. The robot stages consist of the position and velocity of the gripper and the robot joints. The goal vector consists of the target position coordinates.

## A.3 ACTION SPACE

For robot tasks, the action is a 8-dimensional vector, which is the concatenation of two 4-dimensional action vectors for each robot. For each robot, the first 3 elements indicates the desired position shift of the end-effector and the last element controls the gripper fingers (locked in push with door scenario). For mass point tasks, the action is a 4-dimensional vector, similarly combined with two 2-dimensional vectors for each mass point. The 2-dimensional action vector only controls the the desired position shift of the end-effector in a plane.

## B TRAINING SAMPLE NUMBER

For *Push with Door* and *Open Box and Place*, we use 10M samples to train; For *Tower Stack* and *Rearrangement*, we leverage curriculum learning and increase the number of blocks by one in each stage. Specifically, we list the number of samples for each stage of training in Table 6.

Table 6: Training samples for curriculum learning in each stage

| Num of Block | 1 | 2 | 3 |
|---|---|---|---|
| Tower Stack | $2 \times 10^6$ | $6 \times 10^6$ | $9 \times 10^6$ |
| Rearrangement | $1 \times 10^6$ | $3 \times 10^6$ | $5 \times 10^6$ |

**Computation.** In our experiments, we use a single GPU and 8 CPU cores for all the method on each task.

## C VIDEO RESULTS

Please refer to our project page: https://bimanual-attention.github.io/.

