# OpenReview forum: "DAIR: Disentangled Attention Intrinsic Regularization for Safe and Efficient Bimanual Manipulation"
_ICLR.cc/2022/Conference — ICLR 2022 Submitted_

### Official Review · Reviewer_v16H · 2021-11-01

**Correctness:** 3
**Technical Novelty And Significance:** 2
**Empirical Novelty And Significance:** 2
**Recommendation:** 6
**Confidence:** 3

**Main Review:**

The paper is written in a clear way, it is well motivated and it clearly defines the domain challenges that are going to be addressed. The proposed method is formulated in a fairly clear way (see comments below). The experimental results are carried out on a variety of tasks involving the two robot’s collaboration.
 Comments:
- Please clarify earlier (e.g. in “Method”) what the state of the robot arms is, as this is not clear until much later in the paper. This information could help understanding the method better from the beginning
- How is an “interaction region” (mentioned in “Network architecture”) defined? Is it defined based on the robots’ states in simulation? How would it be defined in a real world scenario?
- Regarding state encoders and agents: what do f_{i,j}(.) encode exactly? As far as I understand it, f_{i,j} takes state s_j as input and outputs an encoded vector, “according to” agent i? What is the meaning/role of “agent” in this context?
- The previous point on state encoders and agents is not made clear in Fig. 2 either: here, all encoders have the subscript “i”, does it mean that there is only an agent “i”? Also, what is s_i with respect to s_1, s_2, etc.? Also, is “interaction regions” equivalent to an agent (ref. Fig. 2)?
- In paragraph “Implementation details”, what is “target interacting position, velocity, …” referring to? Do they indicate the target position of the two robots + objects?
- In your experimental results sections, can you specify how many repetitions you performed to obtain the scores reported in the tables? Several times the std (?) of the scores reported is quite high with respect to the means, why is that the case? How do you account for that?
- Success rates for adaptation stages in Table 3, even though better than the Attention scores, show a significant drop in performance with respect to the learning cases. Can you discuss this drop? How could this be improved?
- Fig. 7 is missing a legend for the two curves depicted.


**Summary Of The Paper:**

This paper addresses important aspects of bimanual manipulation, in particular domination and conflict, by proposing a technique called disentangled attentions, which provides an intrinsic regularization for two robots to focus on separate subtasks/objects. The proposed method is evaluated in simulation, using two robot arms to solve a variety of tasks on table-top environments.


**Summary Of The Review:**

The paper presents a well formed idea and method, but has some points to be clarified that would improve the manuscript.

---

### Official Review · Reviewer_RR63 · 2021-11-01

**Correctness:** 3
**Technical Novelty And Significance:** 3
**Empirical Novelty And Significance:** 2
**Recommendation:** 5
**Confidence:** 3

**Main Review:**

## Strengths and Weaknesses
### Things I liked about this work:
- **an interesting idea**: I really like the idea of intrinsic regularisation by using attention. It seems like a useful technique that may lead to exciting future work too.
- **a well structured paper**: The overall narrative and flow of the paper is very well put together so it is easy to read and understand.
- **a good set of experiments**: I also like the type and number of experiments ran and their overall presentation.

### Things that can be improved:
- **misleading contribution**: there are a few issues with the current state of the claimed contribution. Claiming as contribution the observation that dominance and collision can occur in dual-arm control is misleading. These are not novel observations and are therefore not a contribution to this work. In addition, the claim that the solution ensures preservation of safety in collaborative shared spaces is also a bit misleading because it is implicitly learnt over trial and error with no follow up guarantees. See below for more details.
- **insufficient number of seeds - 3 is not enough**: although the paper does a great job at evaluating different aspects of this work, it uses 3 seeds per experiment which is significantly insufficient for model-free RL, especially when there are complex dynamics introduced from the manipulation set up and the bimanual nature of the tasks. I find this the biggest weakness of this paper.
- **overall clarity can be further improved**: although the paper does a great job at explaining the methodology and presenting the achieved results, there are a number of unclear statements that can be further improved.

## Claim and Contribution
One concern I have for this work is the practicality of the solution. A large part of the claim is on preserving safety in collaboration in shared spaces. This is truly an important problem in physical systems although not necessarily a big deal for simulation. Although there is nothing wrong with proposing a solution to this with simulated evaluation, it still seems important to somehow ground it to the context where it practically matters. However, as it stands, it is hard to see how this solution can be at all scaled in practice. Specifically, the approach only implicitly stimulates reduced collision through regularisation. This, however, does not prevent from collision in the process of learning and what is worse is that it does not provide any guarantees that there will be no collision once a policy has been extracted. This seems to deem the solution impractical for part of its claimed contribution. However, if there is another benefit to minimising collision in the context of the simulated task and collaborating agents then this should be explained in more details. Otherwise, the claim seems a bit conflicting.

In addition, the way the work by Chitins et al. is introduced makes me think this work is directly related to this problem. 'While this intrinsic reward encourages the two robots to collaborate for tasks that are hard to achieve by a single robot, it does not address the domination and conflict problems for efficient and safe manipulation.' Encouraging robots to collaborate seems directly related to addressing the domination problem (e.g. to not collaborate) so this seems like a direct candidate to compare against. If the methods are not comparable for some reason, then its reasons should be explained clearly.

The concept of robot collaboration describes the act of interacting agents and is thus inherently solving the problem of using a dominant agent, e.g. through coordination [1,2,6,7] for example. There is a lot of work in robotics done on safe collaboration in bimanual settings too [3,4,5] too, so it is not clear why this is a contribution for this work. Moreover, those works were not part of the literature review but perhaps some of them might help put things in perspective.

Furthermore, the summarised contribution on page 2, states that the proposed approach can solve the tasks more efficiently but this is not necessarily true when looking at the evaluation section. Requiring 10M as opposed to 11M environmental steps, for a 100 step long horizons, is hardly more efficient. Perhaps the solution results in faster learning but it does not seem more efficient.

## Experimental Evaluation
I think this work does a great job at presenting the overall experimentation. I really like the breadth of evaluating the work but I have a few additional concerns. The main one is the small number of seeds used. To be able to confirm the utility of this method, I think it's important to run experiments with at least 10 seeds. Being a model-free approach, it is well known problem that the learning of a Q function can vary drastically even if the tasks were not manipulation heavy and not bimanual. The problem of interest in this work asks for more thorough evaluation with respect to the number of seeds. I would happily reconsider my recommendation if this gets fixed.

In addition, it seems like the proposed approach works better with a fully sparse reward as opposed to an informative one, is there a reason for this, it seems somewhat counter intuitive.

The overall study of the required finish steps is great but it seems like it is performed on a not-yet converged policies. In this case it is not clear if the reported improvement against the number steps required is really thanks to obtaining a more effective policy, synthesised with the help of the provided regularisation, or is merely a consequence of the not-yet converged policies. This brings me back to the comment made in claims and contribution, perhaps a more accurate statement would be to measure the speed of learning, e.g. as a function of the per-step normalised reward, as opposed to the speed of solving the task. It seems only meaningful to measure the speed of solving a task between two learning processes that have already converged to near-optimal policies. This is not the case according to Figures 3 and 4.

## Minor comments
'One possible solution is to design a task-allocation reward function to encourage better coordination.' seems like a statement that needs a reference.

## References

[1] Lee, Y. et al  Learning to coordinate manipulation skills via skill behavior diversification. ICLR 2019

[2] Hu, B. et al. Coordinated compliance control of dual-arm robot astronaut for payload operation. IJARS 2021

[3] Sadeghian, H. et al 2012. Global impedance control of dual-arm manipulation for safe interaction. _IFAC Proceedings Volumes_, _45_(22), pp.767-772.

[4] Vick, A. et al. Safe physical human-robot interaction with industrial dual-arm robots. In _9th International Workshop on Robot Motion and Control_ (pp. 264-269). IEEE.

[5] Gams, A. et al. Coupling movement primitives: Interaction with the environment and bimanual tasks. 2014 _IEEE Transactions on Robotics_, _30_(4), pp.816-830.

[6] Deng, M. et al, 2017, August. Reinforcement learning of dual-arm cooperation for a mobile manipulator with sequences of dynamical movement primitives. In _2017 2nd International Conference on Advanced Robotics and Mechatronics (ICARM)_ (pp. 196-201). IEEE.

[7] Freek Stulp et al. Reinforcement learning with sequences of motion primitives for robust manipulation. IEEE Transactions on robotics, 28(6):1360–1370, 2012.


**Summary Of The Paper:**

This paper proposes an attention-based solution to dual-arm robot manipulation from sparse rewards that relies on a novel idea for intrinsic regularisation. The proposed regularisation term encourages each robotic arm to focus on separate subtasks and objects. The proposed approach aims to reduce the problem of extracting a dominating agent in collaborative settings and to reduce the number of collisions between operating robots in a shared workspace. This work is evaluated in simulation and the obtained results demonstrate the ability of the proposed solution, DAIR, to not only improve both the success rate and sample efficiency of the learning process but also to reduce the number of conflicts between the two operating arms. The approach is interesting and the result seem promising but I have some concerns and additional questions that I detail below.

**Summary Of The Review:**

Overall the paper introduces an interesting and simple solution to a complex bimanual set up. However, there are some details around the claims and contributions that appear not clear from the current state of the paper. The provided evaluation resorts to a very small number of seed runs (3) which makes it hard to asses the significance of the reported results. Therefore, I cannot yet recommend this work for acceptance. I would happily reconsider my score if the authors can clarify the detailed above issues and provide a more thorough evaluation in support of their solution.

---

### Official Review · Reviewer_a42g · 2021-11-03

**Correctness:** 3
**Technical Novelty And Significance:** 3
**Empirical Novelty And Significance:** 3
**Recommendation:** 5
**Confidence:** 4

**Main Review:**

### Strengths

- The proposed idea is very intuitive and well-motivated.
- The paper is well written.
- The comparisons and experimental settings are interesting and helpful to understand the proposed *disentangled attention* and *intrinsic regularization*.
- Figure 5 is great to show the disentangled attention learned with the proposed attention model and intrinsic regularization. Can the authors provide the video version of Figure 5 over an episode to make sure it is not cherry-picked? Also, it would be helpful to compare the learned attention with the Attention baseline.
- The generalization capability and collaborative behaviors shown in the experiments are impressive and significantly better than the baseline approaches, which has the potential to extend to more complicated multi-agent RL problems.


### Weaknesses

- The proposed regularization prevents two robots to work nearby each other. Although this can generally prevent collision between two robots, it also limits close collaboration between two robots. It is not clear why the experiment in Table 5 and Figure 7 suggests that DAIR does not prevent synergistic skills. In this experiment, the state of the target object is split into two separate parts so that the intrinsic regularization can work on the same object. To be fair, the results when the state space is not tailored to the proposed method are required. Moreover, the results in Table 5 and Figure 7 show that the Attention baseline is comparable to the proposed method (the error intervals are overlapping) even if the experiment is designed prone to the proposed method.
- The difference between adding collision penalty ("Attention + collision penalty") and adding intrinsic regularization ("DAIR") seem not drastically different in terms of the final conflict rate. Given that both approaches require task-specific domain knowledge and the performance difference becomes not significant, it is debatable which one is better. A more interesting question would be whether this method can help reduce collision even during training, not just for the final conflict rate.
- The Attention baseline shows significantly lower performances in Table 3 and Table 4. Is this mainly due to the conflict between two arms? or more due to the regularization effect for scaling? If the former is the case, comparing it with the "Attention + collision penalty" baseline can be helpful to support the claim of the paper in the implicit regularization.
- The paper points out that the generalization capability of the proposed method may come from regularizing parameters not to overfit. The explanation about the generalization capability of the proposed method can be more thoroughly investigated. Following the suggested reason, will including some regularization techniques (such as spectral norm, L2 norm, mix-up) improve the generalization capability of the Attention baseline? or is the proposed implicit regularization much stronger than these prior regularizations?
- Figure 4 explains that the proposed method is robust to the choice of $\lambda$, but models with different $\lambda$ show noticeable differences in performance except Push with Door (Informative).
- All experiments are done in the state space, which requires the state space to be hand-engineered for the proposed method. It would be interesting to see that it works with more complicated high-dimensional inputs, such as the visual domain, where the split of the state space is not available.
- Throughout the paper, one of the main claims is that the proposed method can resolve the "domination" issue. However, it is not intuitive how the proposed method resolves the problem of domination.
- It is unclear how HER is working with the proposed model, especially with rewards. Is HER only used for Stack Tower and Rearrangement tasks? Also, is there any synergetic effect by using HER together with the proposed method, or can the proposed method be used alone without HER in other applications? Clarification on HER is required.


### Minor comments

- In the preliminary section, $\pi$ is parameterized by $\theta_i$ but it should be $\phi_i$ to be consistent with Section 3.1.
- Citation and comparisons missing for a closely related work
  - Lee et al. Learning to Coordinate Manipulation Skills via Skill Behavior Diversification, ICLR 2020
  - Tung et al. Learning Multi-Arm Manipulation Through Collaborative Teleoperation, ICRA 2021
- The maximum episode length is specified as 100 in Section 3.2 while it depends on the number of objects in Section 5.
- Given the high variance in the proposed method and comparable performance of the "Attention" baseline, it would be more convincing to have more random seeds.

**Summary Of The Paper:**

This paper proposes an implicit regularization for bimanual manipulation that enforces two robot arms to focus on different regions, which prevents both arms from performing on the same object at the same time. The proposed method realizes this idea by computing attention between robots and objects and then constraining dot product between attentions of two arms to not overlap. This effectively prevents the conflict between two arms by encouraging two arms to focus on different objects (sub-tasks), leading to efficient RL training and safe behaviors. The empirical evaluation demonstrates that the proposed method is generalizable to unseen situations thanks to the attention mechanism with the regularization.

**Summary Of The Review:**

This paper proposes a novel inductive bias (the attention network and regularization) for efficient bimanual manipulation, which outperforms the simple MLP and multi-agent baselines. The paper is easy to follow and experiments are well explained. However, there are certain limitations in observation spaces and applications. In addition, analysis of some experiments is not convincing (e.g. generalization, synergistic interactions) and it needs additional experiments to support the claims of the paper. I would like to see the authors' responses and adjust my recommendation accordingly.

---

> ### Comment · Reviewer_a42g · 2021-11-29
> **After rebuttal period**
>
> Given the missing author response and negative reviews across the reviewers, I would maintain my current rating, weak rejection.

---

### Official Review · Reviewer_jKrv · 2021-11-05

**Correctness:** 2
**Technical Novelty And Significance:** 3
**Empirical Novelty And Significance:** 2
**Recommendation:** 3
**Confidence:** 5

**Main Review:**

The paper presents a simple method for regularising the training of a multi-agent system. An attention mechanism serves as a method for disentangling the representations for each agent helping them to attend their respective task, without intervening with the actions of the other agent, thanks to the regularisation term. I find the idea simple and intuitive. I have strong concerns regarding the plenty assumptions being made, in particular for the pre-computed interaction areas, and the scalability of the method to more than two agents. The authors falsely abuse the term bimanual manipulation. The presented experimental setting considers two agents that collaborate, and they happen to be manipulators. The problem at hand falls into then category of collabarotive multi-agent systems, and not bimanual manipulation. A bimanual manipulator is a single agent with two arms, that observes the same state and has to coordinate both arms for achieving a goal. While it may sound the same, the experimental setting and the whole motivation of this paper is wrong. Especially, because I do not see any reason why the system could not scale to three or four robots. Such overstatements are very problematic, especially because the authors refer to a machine learning conference, and they are transferring false interpretations about problems that have been long studied in robotics. Apart from that the authors abuse the term safety. There is no connection of this work with the broad literature of safe learning, safe RL or safe planning. The fact that you are measuring number of collisions and you are getting some better results do not provide any formal guarantees about safety. At most, you can state that the method is more efficient than a baseline when the number of collisions is lower.

In Section 3.1 the authors falsely claim that in SAC you have to learn the temperature parameter; the temperature parameter is not learned but is optimised throughout the learning, till a target entropy is reached. In section 4.1, the authors state “ For Q-function, there are two modifications: (i) The state encoder $f^Q(s_i , a_i)$ for the agent i also takes in the action as inputs; (ii) The final layer of i,ithe Q-function network outputs a scalar value instead of an action distribution as $Q^{θ_i}(s^t , a^t_i )$, which is used in Eq. 1 and Eq 3”. This statement is completely wrong. SAC takes as input states and actions and already outputs a scalar. It seems to me that the authors are confusing SAC with DQN, and they have not sufficient experience with SAC.

For the experimental evaluation, the authors do not compare to the most recent Multi-agent RL methods, like MAAC (http://proceedings.mlr.press/v97/iqbal19a/iqbal19a.pdf). The authors seem to also be using HER for a goal-based RL setting. Are you doing the same implementation for MADDPG, i.e., implementing as MAHER? Moreover, the authors state that they tried to implement MASAC but the results are the same. That is hard to believe, so I request that the authors append the results in the supplementary. The results are not convincing, as they are not statistically significant. This is also evident by the high variance in the learning curves, and the overlaps between the simpler Attention baseline and DAIR. The results in Table 1 do not convince me in particular for the Door (sparse) as from Figure 3 the two compared methods have the same performance. How do you justify this?
In Section 5.3, the authors overstate the performance of DAIR when clearly Figure 7, does not show any statistically significant results.

Overall, the evaluation is lacking both for the compared algorithms and for the reliability of the results. I would also recommend that the authors present different multiagent experimental settings, e.g., like the ones in the MADDPG paper, with more than one papers to showcase the representation power of the proposed attention mechanism, and necessarily provide comparisons against the aforementioned method MAAC.


**Summary Of The Paper:**

The paper presents an attention mechanism that also serves as intrinsic regularisation for a two-agent reinforcement learning framework of collaborative tasks. The method is evaluated in simulation using two manipulator robots, and has been compared against a simpler version of the proposed method (attention without regularisation) and a standard multi-agent RL method.

**Summary Of The Review:**

The paper provides a very simple, yet seemingly effective approach. They do not provide statistically significant results, as they only evaluated with 3 seeds, and this is evident in the results. Moreover, they do not compare with all possible baselines on the topic. Importantly, the paper reproduces false claims, abusing the term bimanual manipulation and the term safety, showing that the authors clearly are not in touch with the topic of robotic manipulation. If the authors change their narrative and fix their motivation, provide sufficiently more seeds to make their results reliable, and also compare with appropriate baseline, I might consider increasing my score,

---

### Decision · Program_Chairs · 2022-01-20

**Decision:**

Reject

**Comment:**

The paper presents a method for collaborative task solving via an attention mechanism. The method is evaluated on manipulation task in simulation.

The reviewers agree that the paper is well written and the idea is novel and intuitive. They also share concerns about the limited applicability (too many assumptions and only two robots) of the method and that it contains unjustified claims, and therefore does not meet the ICLR bar.

Constructive feedback for the next version of the manuscript:

- The authors should decide if this is a robot learning (a learning-based method that advances robotics, specifically robot manipulation) or a machine learning paper (a method that advances cooperative multi-agent learning). The decision should drive the publication venue and the baselines. If the target is robot learning, the paper should consider adding on-robot experiments. If the target is ML method, that more baselines and benchmarks from the MARL community should be added to the evaluation section.
- The authors should be careful not to confuse safety guarantees, which have theoretical and analytical implications, with empirical evaluation without collisions.
- Evaluate the learning on more that 3 seeds.